# Structure guided mimicry of an essential *P. falciparum* receptor-ligand complex enhances cross neutralizing antibodies

Sean Yanik[1,2], Varsha Venkatesh[1,2], Michelle L. Parker[3], Raghavendran Ramaswamy[3], Ababacar Diouf[4], Deepti Sarkar[1,2], Kazutoyo Miura ®[4], Carole A. Long ®[4], Martin J. Boulanger ®[3] & Prakash Srinivasan ®[1,2] ✉

Invasion of human erythrocytes by *Plasmodium falciparum* (*Pf*) merozoites relies on the interaction between two parasite proteins: apical membrane antigen 1 (AMA1) and rhoptry neck protein 2 (RON2). While antibodies to AMA1 provide limited protection against *Pf* in non-human primate malaria models, clinical trials using recombinant AMA1 alone (apoAMA1) yielded no protection due to insufficient functional antibodies. Immunization with AMA1 bound to RON2L, a 49-amino acid peptide from its ligand RON2, has shown superior protection by increasing the proportion of neutralizing antibodies. However, this approach relies on the formation of a complex in solution between the two vaccine components. To advance vaccine development, here we engineered chimeric antigens by replacing the AMA1 DII loop, displaced upon ligand binding, with RON2L. Structural analysis confirmed that the fusion chimera (Fusion-F$_{D12}$) closely mimics the binary AMA1-RON2L complex. Immunization studies in female rats demonstrated that Fusion-F$_{D12}$ immune sera, but not purified IgG, neutralized vaccine-type parasites more efficiently compared to apoAMA1, despite lower overall anti-AMA1 titers. Interestingly, Fusion-F$_{D12}$ immunization enhanced antibodies targeting conserved epitopes on AMA1, leading to increased neutralization of non-vaccine type parasites. Identifying these cross-neutralizing antibody epitopes holds promise for developing an effective, strain-transcending malaria vaccine.

Malaria caused by *P. falciparum* remains an immense global health and economic concern and is responsible for the majority of the 627,000 malaria-related deaths in 2021[1]. Merozoite invasion of RBCs can be considered the gateway to disease as it is the parasites growing within the safety of the host cell that causes clinical symptoms in susceptible individuals. RTS,S, the first WHO authorized malaria vaccine targeting the clinically silent forms of the parasite has

limited efficacy and there is a growing concern due to the development of resistance to frontline antimalarials[2,3]. There is an urgent need for a vaccine that can reduce the parasite burden in the blood and prevent disease. People living in endemic countries, who are exposed to repeated malaria infections, can develop clinical immunity[4]. AMA1 is among the most immunogenic parasite targets, and anti-AMA1 antibodies inhibit merozoite invasion[5,6]. AMA1

[1]Department of Molecular Microbiology and Immunology, Johns Hopkins School of Public Health, Baltimore, MD 21205, USA. [2]The Johns Hopkins Malaria Research Institute, Baltimore, MD 21205, USA. [3]Department of Biochemistry and Microbiology, University of Victoria, Victoria, BC V8W 3P6, Canada. [4]Laboratory of Malaria and Vector Research, National Institutes of Allergy and Infectious Diseases, National Institutes of Health, Rockville, MD 20852, USA. ✉e-mail: psriniv3@jhu.edu

function is critical for both merozoites and sporozoites, and in *P. falciparum*, its interaction with RON2 is required for invasion[7–10]. Positive selection of polymorphisms in AMA1, particularly in regions surrounding the RON2 binding site, suggests it is an important target for neutralizing antibodies[11]. However, AMA1 vaccines in phase 2 clinical trials failed to protect against vaccine-type parasites despite generating high antibody titers[12–15], suggesting that the vaccine did not induce sufficient neutralizing antibodies.

Antigen redesign to focus the immune response to critical epitopes may help to enhance the proportion of neutralizing antibodies induced by the vaccine. Previous studies using an AMA1-RON2L binary complex demonstrated greater protection than apoAMA1 against *P.yoelli* in a mouse model[16] and *P. falciparum* in a non-human primate malaria model[17]. Vaccine efficacy was strongly correlated with the ability of the binary complex vaccine to increase the proportion of neutralizing antibodies targeting AMA1-RON2 interaction[16,17]. This enhancement in neutralizing antibodies was not only limited to vaccine-type parasites but also against some heterologous parasites[16,17]. Despite these encouraging results, manufacturing and deploying a vaccine that relies on producing and mixing two proteins that need to spontaneously assemble in solution, presents technical challenges. To facilitate vaccine development, we engineered a single chimeric antigen that would recapitulate the receptor-ligand complex and promote the effective development of neutralizing antibodies against *P. falciparum*.

## Results

### Engineering a receptor-ligand fusion chimeric malaria vaccine

AMA1 is comprised of three domains (Fig. 1A) with domains 1 and 2 together forming a hydrophobic groove, the binding site for RON2L[18]. In this study, we generated fusion chimeras to mimic the structure of the receptor-ligand complex in a single protein immunogen. We replaced a section of the extended PfAMA1 DII loop close to the RON2L binding site that is largely disordered in the apo structure[18] (AMA1 residues T358-K370), with RON2L (RON2 residues T2023-S2059) (Fig. 2A; Suppl Figs. 1 and 2G). We initially generated two recombinant chimeras with the RON2L sequence positioned either in the same direction as the AMA1 primary sequence (Fusion-$F_{D123}$) or in the reverse direction (Fusion-$R_{D123}$) (Suppl Fig. 1A). However, recombinant production of these three-domain chimeras ($AMA1_{D123}$) in Sf9 cells proved unsuccessful. Previous studies showed that binding of a *Toxoplasma gondii* RON2L to its AMA1 partner led to allosteric structural changes in domain 3 of TgAMA1[19]. Such conformational changes in PfAMA1$_{D123}$ may result in protein instability that is not tolerated in the Sf9 heterologous expression system.

### AMA1 domains 1 and 2 are the main targets of AMA1 + RON2L binary complex-induced neutralizing antibodies

As AMA1 domains 1 and 2 are sufficient for RON2L binding[20], we hypothesized that these regions may be sufficient to generate recombinant chimeras that mimic the structure of AMA1 in the

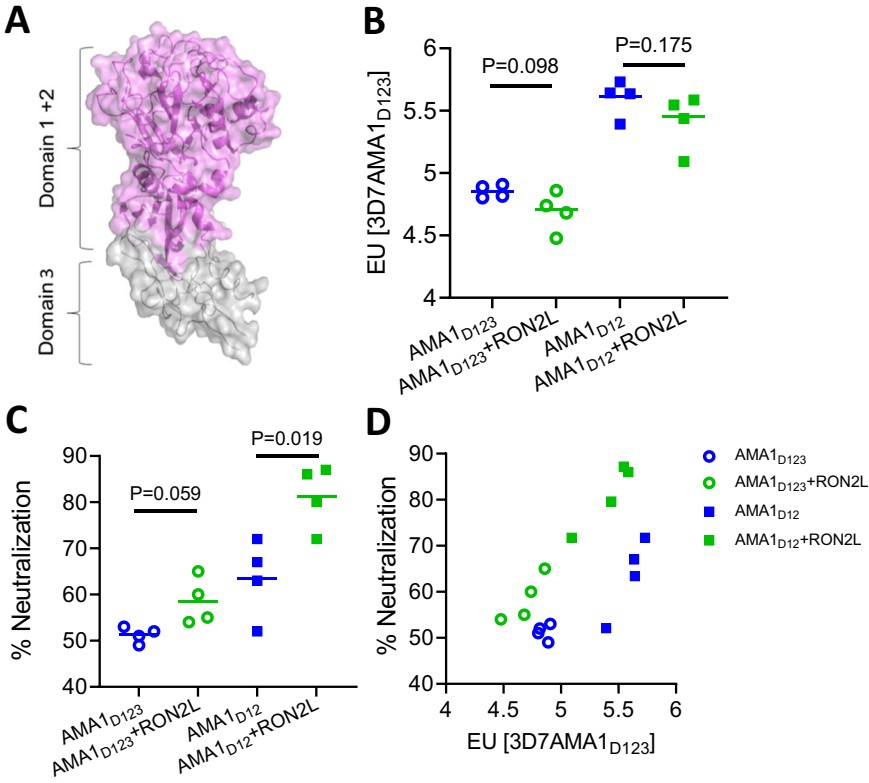

**Fig. 1 | AMA1 domains 1 and 2 are sufficient for complex-mediated enhancement in antibody quality. A** Schematic of AMA1 showing domains 1, 2 (purple) and 3 (gray). Domains 1 and 2 (purple) of AMA1 together form a hydrophobic groove, the binding site for RONL2. Domain 3 (gray) is modeled on PvAMA1 domain 3 (PDB: 1W8K). **B** AMA1-specific antibody titer in the purified IgG from apoAMA1 (blue) and AMA1 + RON2L binary complex (green) immunized rats. Open circle and squares indicate the groups that used $AMA1_{D123}$ (all three domains) and $AMA1_{D12}$ (domains 1 and 2), respectively. Data are presented for individual animals (*n* = 4 per group) and each data point is the average of three replicates. Horizontal lines show the mean titer in each group. Two-tailed Welch's *t*-test was performed to compare differences between groups. **C** In vitro neutralization (1-cycle) assay against vaccine-type 3D7 parasites using 2 mg/mL of total IgG from each animal. Data are from individual animals (*n* = 4 per group) and each data point is the average of three replicates. Horizontal lines show the mean neutralizing activity in each group. Two-tailed Welch's *t*-test was performed to compare differences between groups. **D** Relationship between anti-AMA1 titer in the IgG (*x*-axis) and neutralizing (1-cycle) activity in 2 mg/mL total IgG (*y*-axis).

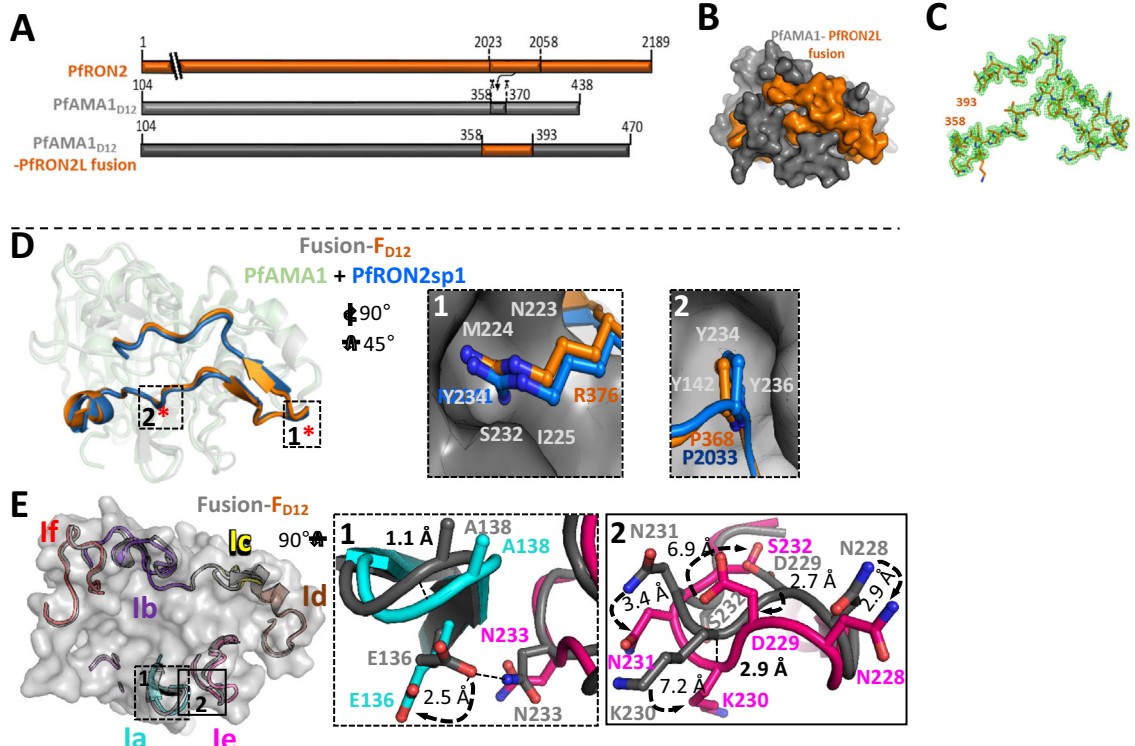

**Fig. 2 | Engineered chimera mimics *P. falciparum* receptor-ligand binary complex. A** Schematic showing the region of AMA1 DII loop that was replaced with RON2L. **B** Surface view of PfAMA1-RON2L fusion with the same color code as shown in **A**. **C** 2Fo-Fc electron density map for *Pf*RON2L contoured at 1.0 σ, highlighting well-ordered density from the N-terminus (residue 358) to the C-terminus (residue 393). **D** Structural overlay of PfAMA1-PfRON2 binary complex (PDB: 3ZWZ) with Fusion-$F_{D12}$. Color scheme is same as in A. The key interactions driving the binding of PfRON2L to PfAMA1 in the binary complex indicated by asterisks (*) are conserved in Fusion-$F_{D12}$. Box 1 shows R376 in the chimera (R2041 of PfRON2) fits snugly into the pocket of PfAMA1 stabilized by hydrogen bonds. Box 2 shows critical residues (P368/R2033) that form identical interactions with PfAMA1.

**E** Structure overlay of binary complex and engineered chimera Fusion-$F_{D12}$ showing AMA1 loops surrounding the RON2L binding site. The loops of the Fusion-$F_{D12}$ chimera are colored and the corresponding loops in binary complex are shown in gray. Box 1 shows the loop 1a and a part of loop 1e that interacts with loop 1a. A hydrogen bond formed between N233 (loop 1e) and E136 (loop 1a) in the binary complex structure that is lost in the chimera. The curved arrows indicate the displacement of the sidechains. Box 2 shows the differences in loop 1e. There is a displacement of ~2.9 Å between the loops, most likely caused by the inability to form hydrogen bond between N233-E136 in chimera. The different orientations of the sidechains are indicated by curved arrows and the extent of displacement is indicated.

binary complex. Since earlier vaccine studies used AMA1$_{D123}$ containing all three domains to generate the binary complex[16,17] and some inhibitory antibodies target domain 3[21,22], we first tested if domain 3 is required for the enhancement in antibody quality generated by the binary complex vaccine. Mean AMA1-specific antibody titers from rats immunized with apoAMA1 (using AMA1$_{D123}$ or AMA1$_{D12}$) was slightly higher compared to rats immunized with the corresponding binary complex (AMA1$_{D123}$ + RON2L or AMA1$_{D12}$ + RON2L) antigens (Fig. 1B). Despite this, IgG from the binary complex groups neutralized vaccine type parasites (Pf3D7 strain) to a greater degree than the corresponding apoAMA1 groups (Fig. 1C). Furthermore, comparing anti-AMA1 titer of IgG from rats immunized with apoAMA1$_{D12}$ and AMA1 + RON2L binary complex revealed that at similar antibody titer IgG from the binary complex immunized animals exhibited greater parasite neutralization than the apoAMA1 groups (Fig. 1D), indicating that the binary complex immunogen generated better quality antibodies. Importantly, these results show that a receptor-ligand binary complex containing only domains 1 and 2 of AMA1 (AMA1$_{D12}$) is sufficient to improve vaccine quality. It is also noteworthy that at the same immunogen dose used for vaccination (10 µg/rat), the AMA1$_{D12}$ + RON2L binary complex generated higher levels of neutralizing antibodies compared to AMA1$_{D123}$ + RON2L (Fig. 1D), suggesting that AMA1 domains 1 and 2 contain the key targets of neutralizing antibodies.

## Fusion chimera closely mirrors the structure of the AMA1 + RON2L binary complex

Two recombinant chimeras, Fusion-$F_{D12}$ and Fusion-$R_{D12}$, each on two different AMA1 allele backbones (3D7 and HP41) and both lacking domain 3 (Suppl Fig. 1B), were expressed in Sf9 cells and verified using conformation dependent mAbs[11,23] (Suppl Fig. 2A, D). We hypothesized that if RON2L bound to AMA1$_{D12}$ as designed, then the hydrophobic binding groove would be occupied and no longer available to bind free RON2L in solution. Enzyme-linked immunosorbent assays (ELISAs) showed that free RON2L readily bound to apoAMA1$_{D12}$ and Fusion-$R_{D12}$ but not Fusion-$F_{D12}$ chimera (Suppl Fig. 2E, F). This suggests that the hydrophobic binding groove in AMA1$_{D12}$ is occupied by RON2L in the forward (Fusion-$F_{D12}$) but not in the reverse (Fusion-$R_{D12}$) chimera. We next sought to establish a detailed molecular blueprint of the Fusion-$F_{D12}$ chimera using X-ray crystallography. A fusion chimera engineered using the HP41 AMA1 sequence was ultimately selected for structural studies based on yield and stability of the recombinant protein.

The structure of the Fusion-$F_{D12}$ protein incorporating residues 105 (2nd residue in the chimera) through 470 including the inserted PfRON2L sequence was determined to 1.55 Å resolution (Fig. 2B). Clear, contiguous electron density in the apical groove of Fusion-$F_{D12}$ enabled unambiguous tracing of the entire PfRON2L sequence (Fig. 2C). Structural analysis revealed that the Fusion-$F_{D12}$ protein adopts a conserved architecture relative to the structure of the previously observed binary complex[20] with a root mean square deviation

(rmsd) of 0.61 Å over 252 Cα positions. Moreover, RON2L in the fusion and binary complex overlay with an rmsd of 0.53 Å over 34 Cα positions (Fig. 2D). The core interactions between RON2L and AMA1 are retained in Fusion-$F_{D12}$ including R376 of PfRON2L (R2041 in PfRON2) that docks into a pocket at the end of the PfAMA1 apical groove (Fig. 2D, Box 1), and P368 (P2033 in PfRON2) that docks into a pocket formed by Y142, Y234, and Y236 (Fig. 2D, Box 2). Overall, all the AMA1 loops (1b, c, d, and f) that pack against the C-terminal end of the RON2L sequence are highly conserved (rmsd of 0.21 Å over 76 Cα positions) relative to the binary complex. In contrast, some differences are observed in loops 1e/1a that pack against the N-terminal section of the RON2 sequence (Fig. 2E). Notably, the sidechain of E136 on loop 1a is displaced by 2.5 Å and no longer forms a hydrogen bond with the sidechain of N233 positioned on loop 1e (Fig. 2E-Box 1). The loss of the hydrogen bond may also be responsible for the observed loop displacements, though it should be noted that differences in crystal packing between the fusion structure presented herein and the binary PfAMA1-RON2 structure may also contribute to observed shifts. Furthermore, many of the sidechains on loop 1e adopt different rotameric states with N228, N231 and D229 showing relatively similar displacement (-2.9 Å, 3.1 Å, and 3.4 Å) while K230 (-7.2 Å) and S232 (-6.9 Å) show significantly larger shifts (Fig. 2E-Box 2), which may alter epitopes in loop 1e. But overall the interface between PfAMA1 and the inserted

PfRON2L sequence in the chimera appears to faithfully mimic that observed in the binary complex.

## Antigen and adjuvant effects on immunogenicity, IgG specificity, and parasite neutralizing activity

As Fusion-$F_{D12}$ closely mimics the structure of the receptor-ligand binary complex, we next tested the immunogenicity of the chimera to evaluate if antibody quality will similarly be improved compared to $AMA1_{D12}$. We used the recombinant chimeric antigen based on the $3D7AMA1_{D12}$ backbone (Suppl Fig. 1B) as the 3D7 parasite strain is commonly used in neutralization assays. Two different adjuvants were used, AddaVax, a squalene-based oil-in-water nanoemulsion adjuvant whose composition is similar to MF59 used in certain Flu vaccines[24], and Freund's, a water-in-oil emulsion adjuvant known to induce robust antibody responses in rodents (Suppl Fig. 3A, B). Mean antibody titers in the purified IgG (at 10 mg/mL) were in general higher in the Freund's adjuvant group compared to AddaVax group and in the serum for the Fusion-$F_{D12}$ group, though not significant (Suppl Fig. 3C, D). Furthermore, purified IgG from Fusion-$F_{D12}$ group had lower mean AMA1-specific antibody titer compared to apo$AMA1_{D12}$ immunized groups in both adjuvants (Fig. 3A and Suppl Fig. 4B).

Surprisingly, in contrast to the enhancement of the neutralizing activity observed using purified IgG from the $AMA1_{D12}$ + RON2L binary

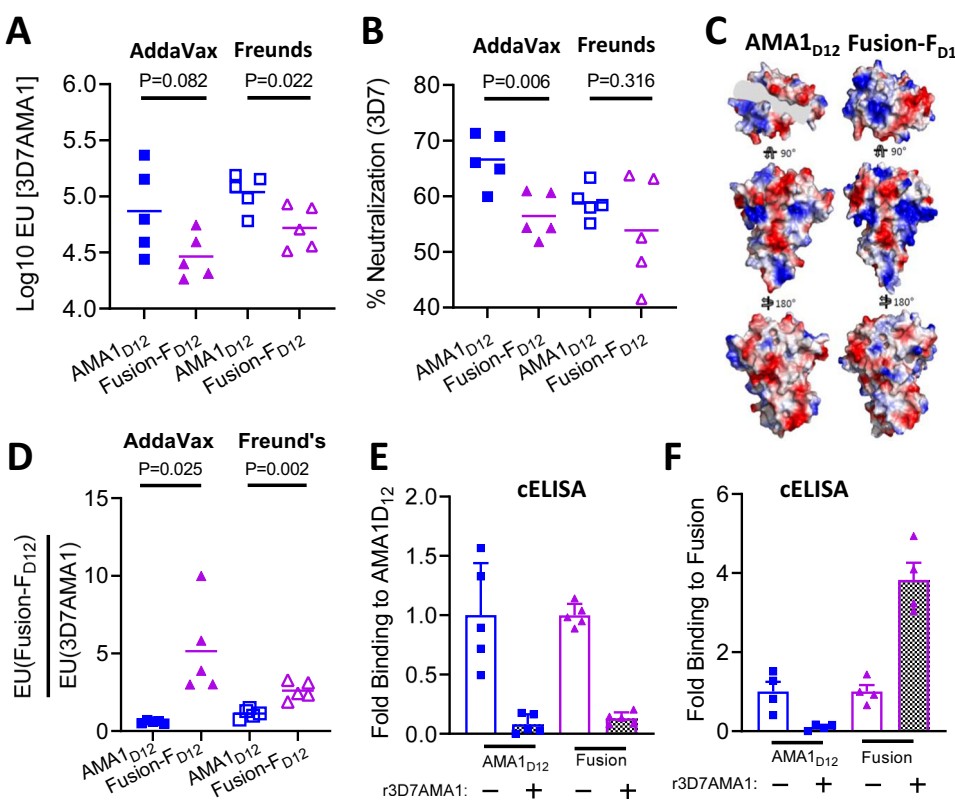

**Fig. 3 | Qualitative changes in vaccine response to apoAMA1 vs. Fusion-$F_{D12}$ immunogens. A** AMA1 titer in purified IgG (10 mg/mL) from animals immunized with apo$AMA1_{D12}$ (blue) or Fusion-$F_{D12}$ (purple) antigens in AddaVax (filled squares and triangles) and Freund's (open squares and triangles). Data are from individual animals (*n* = 5 per group) and each data point is the average of two replicates. Horizontal line marks the mean titer in each group. Two-tailed Welch's *t*-test was performed to compare differences between groups. **B** In vitro neutralization (2-cycle) assay against vaccine-type Pf3D7 parasites. Purified IgG from $AMA1_{D12}$ and Fusion-$F_{D12}$ groups were normalized for 3D7AMA1 titer within each adjuvant group. Data are from individual animals (*n* = 5 per group) and each data point is the average of two replicates. Horizontal line shows mean neutralizing activity in each group. Two-tailed Welch's *t*-test was performed to compare

differences between groups. **C** Differences in surface charge density between apo$AMA1_{D12}$ and Fusion-$F_{D12}$. **D** Proportion of IgG in animals within each group binding to Fusion-$F_{D12}$ and apoAMA1. Data shows ratio of antibody titer from IgG of individual animals (*n* = 5 per group) and each data point is the average of two replicates. Two-tailed Welch's *t*-test was performed to compare differences between groups. **E** Competition ELISA (cELISA) to determine IgG specificity against apoAMA1 antigen in the absence (-) or presence (+) of 2 μM free apoAMA1 between apoAMA1 and Fusion-$F_{D12}$ immunized animals. Assays were performed in duplicate and shown as mean ± SEM (*n* = 5 per group). **F** Competition ELISA (cELISA) to determine IgG specificity against Fusion-$F_{D12}$ antigen in the absence (-) or presence (+) of 2 μM free apoAMA1 between apoAMA1 and Fusion-$F_{D12}$ immunized animals. Assays were performed in duplicate and shown as mean ± SEM (*n* = 5 per group).

complex (Fig. 1C, D and Suppl Fig. 4A), Fusion-F$_{D12}$ immunized animals had lower neutralizing activity (Fig. 3B). It is important to note that the binary complex is a 2-component vaccine in which the proportion of the immunogen in the form of a receptor-ligand complex following immunization is determined by the on-off rate for RON2L binding to AMA1 ($K_D = \sim 20\,nM^{20}$) and may result in a population of dissociated complex in the animal. In contrast, our ELISA results showing that free RON2L peptide is unable to bind to the Fusion-F$_{D12}$ chimera (Suppl Fig. 2E, F) suggests that the binding groove in AMA1 is continuously occupied by RON2L. These apparent differences in RON2L occupancy between the fusion chimera and the AMA1 + RON2L binary complex following immunization may result in the observed differences in anti-AMA1-specific IgG titer and its neutralizing activity. Interestingly, mean anti-AMA1 titer in the purified IgG of the binary complex group was intermediate to that of apoAMA1 and Fusion-F$_{D12}$ groups (Suppl Fig. 4B).

The stable binding of RON2L to the hydrophobic groove of AMA1 in the Fusion-F$_{D12}$ chimera could lead to the generation of new epitopes that are not present on apoAMA1$_{D12}$ resulting in the generation of fusion-specific antibodies. A closer inspection of the surface of the apoAMA1$_{D12}$ and the Fusion-F$_{D12}$ shows distinct differences in charge density particularly within close proximity to the RON2L binding pocket of AMA1 that may influence immunogenicity differently between the two proteins (Fig. 3C). Therefore, while the apoAMA1$_{D12}$ induced antibodies can only target AMA1, the Fusion-F$_{D12}$ immunogen may generate antibodies targeting both AMA1 as well as fusion-specific epitopes, and the overall proportion of such antibodies may impact the neutralizing activity of the polyclonal IgG. Detection of AMA1-RON2 complex-specific antibodies has not been possible previously due to their dynamic interaction. Having a stable fusion chimera allowed us to examine if such antibodies are generated by comparing the ratio of antibodies binding apoAMA1 and Fusion-F$_{D12}$. Animals immunized with the fusion chimera showed a significantly greater ratio of Fusion IgG titer to AMA1 IgG titer, suggesting a proportion of the antibodies induced by the Fusion-F$_{D12}$ may target fusion-specific epitopes (Fig. 3D). As apoAMA1 lacks any fusion-specific epitopes, the increase in the titer ratio in the fusion chimera immunized animals suggests the generation of fusion-specific antibodies. To find direct evidence of the presence of such antibodies we performed a competition ELISA using normalized anti-AMA1-specific IgG from individual animals from each group (Suppl Fig. 6A). As expected, quenching of AMA1-specific antibodies with excess free apoAMA1 abolished reactivity of IgG from the AMA1$_{D12}$ group to both AMA1 and Fusion-F$_{D12}$ antigens (Fig. 3E, F, blue bars). Interestingly, in the fusion chimera immunized animals, IgG binding to Fusion-F$_{D12}$ antigen but not AMA1 was enhanced upon removal of AMA1-specific antibodies (Fig. 3E, F, pink bars). This suggests that AMA1 targeting IgG near the RON2L binding site may compete for binding with fusion epitope-specific antibodies and may influence the overall neutralizing activity of the polyclonal IgG.

## Fusion-F$_{D12}$ enhances the proportion of broadly neutralizing antibodies

In our previous studies, we observed that vaccination with the AMA1 + RON2L binary complex not only enhanced protection against vaccine-type parasites but also increased the proportion of certain strain-transcending antibodies[16,17]. We hypothesized that this is in part due to the binary complex enhancing antibodies against more conserved regions of AMA1 such as loops 1e and 1 f (Fig. 2E). Interestingly, the proportion of antibodies targeting loops 1e and 1f was higher in the Fusion-F$_{D12}$ group compared to apoAMA1$_{D12}$, while no differences were observed against loops 1bcd that contains the highly polymorphic 1d loop (Fig. 4A, C). The higher proportion of loop 1e targeting antibodies in the Fusion-F$_{D12}$ group compared to AMA1$_{D12}$ also suggests that the conformational changes in this loop observed in the fusion chimera

(Fig. 2E, Box 2) did not affect immunogenicity against this important neutralizing antibody target.

Furthermore, the fusion chimera group also had a higher proportion of IgG against AMA1 from the PfFVO strain, a heterologous parasite that differs substantially from vaccine type Pf3D7 AMA1 sequence (Fig. 4D, E; Suppl Figs. 3 and 4C). Interestingly, the proportion of IgG binding to FVO AMA1 (conserved epitopes) in animals immunized with the binary complex (AMA1 + RON2L) was intermediate to that of apoAMA1$_{D12}$ and Fusion-F$_{D12}$ groups thought not statistically significant (Suppl Fig. 4C). Antibodies generated against 3D7AMA1 generally exhibit poor cross-neutralization of heterologous parasites[25]. As the Fusion-F$_{D12}$ group had higher levels of IgG binding to epitopes conserved between 3D7 and FVO strains of *P. falciparum*, we tested if this would differentially affect the invasion of heterologous parasites PfFVO, PfDD2 and Pf7G8 strains. At the same anti-3D7AMA1-specific antibody titer, IgG from Fusion-F$_{D12}$ group neutralized all three heterologous parasites significantly greater than IgG from apoAMA1 (Fig. 4F, H). This suggests that the fusion chimera enhanced antibodies targeting cross-neutralizing epitopes.

## Fusion-F$_{D12}$ immunized rat sera exhibit enhanced parasite neutralization compared to apoAMA1$_{D12}$

While anti-AMA1 antibody titers of purified IgG from the Fusion-F$_{D12}$ group were lower than the apoAMA1$_{D12}$ group, anti-fusion protein titers were proportionally higher in the fusion chimera immunized animals (Suppl Fig. 3A). We reasoned that this difference may be due to the fusion protein preferentially generating antibodies to fusion-specific epitopes that are not present in the apoAMA1 (Fig. 3C). While the serum titer against both apoAMA1 and Fusion-F$_{D12}$ were proportional to the overall immunogenicity within each animal (Fig. 5A, B) there was greater fusion-specific antibodies in the Fusion-F$_{D12}$ group following protein G column purification of IgG (Fig. 5C, D). As these differences were observed using two different adjuvants, it suggested an increase in the proportion of fusion-specific IgG in the Fusion-F$_{D12}$ group after affinity purification. The reason for this enrichment is not clear, however, since differences in antigen specificity of the IgG may influence their biological activity, we compared the antibody titer in the purified IgG and serum for their respective parasite neutralizing activity. At overall similar anti-AMA1 titer, column purified IgG from apoAMA1$_{D12}$ immunized animals had higher neutralizing activity than Fusion-F$_{D12}$ immunized animals (Fig. 5E, F). Surprisingly, however, at the same anti-AMA1 antibody titer, heated inactivated serum from the Fusion-F$_{D12}$ Freund's group neutralized parasites significantly higher compared to the apoAMA1$_{D12}$ group (Fig. 5G and Suppl Fig. 5A), suggesting the Fusion-F$_{D12}$ chimeric antigen enhanced overall antibody quality. The differences in the neutralizing activity of serum compared to purified IgG prompted us to examine potential differences in the immunoglobulin type in the sera of these animals. Generally, a Th1 immune response leads to the production of IgG2a isotype while a Th2 response leads to IgG1 isotype antibodies[26]. Fusion-F$_{D12}$ immunized animals had proportionally higher levels of the IgG1 isotype antibodies (Suppl Fig. 5B, C). We also observed a greater proportion of IgM type antibodies against the fusion chimera in the Fusion-F$_{D12}$ immunized serum compared to apoAMA1$_{D12}$ (Suppl Fig. 5C). Notably, IgM type antibodies against malaria antigens including AMA1 are produced robustly after malaria infection, expand with repeated exposure, and exhibit neutralizing activity[27]. No significant differences in IgA type antibodies were observed between the groups (Suppl Fig. 5E). Additionally, only low levels of RON2L directed antibodies were detected in the Fusion-F$_{D12}$ group (Suppl Fig. 5D), levels that are not likely to influence neutralizing activity[8]. The impact of vaccine-induced differences in Ig type or IgG isotype targeting AMA1 on *P. falciparum* parasites in vivo remains to be determined.

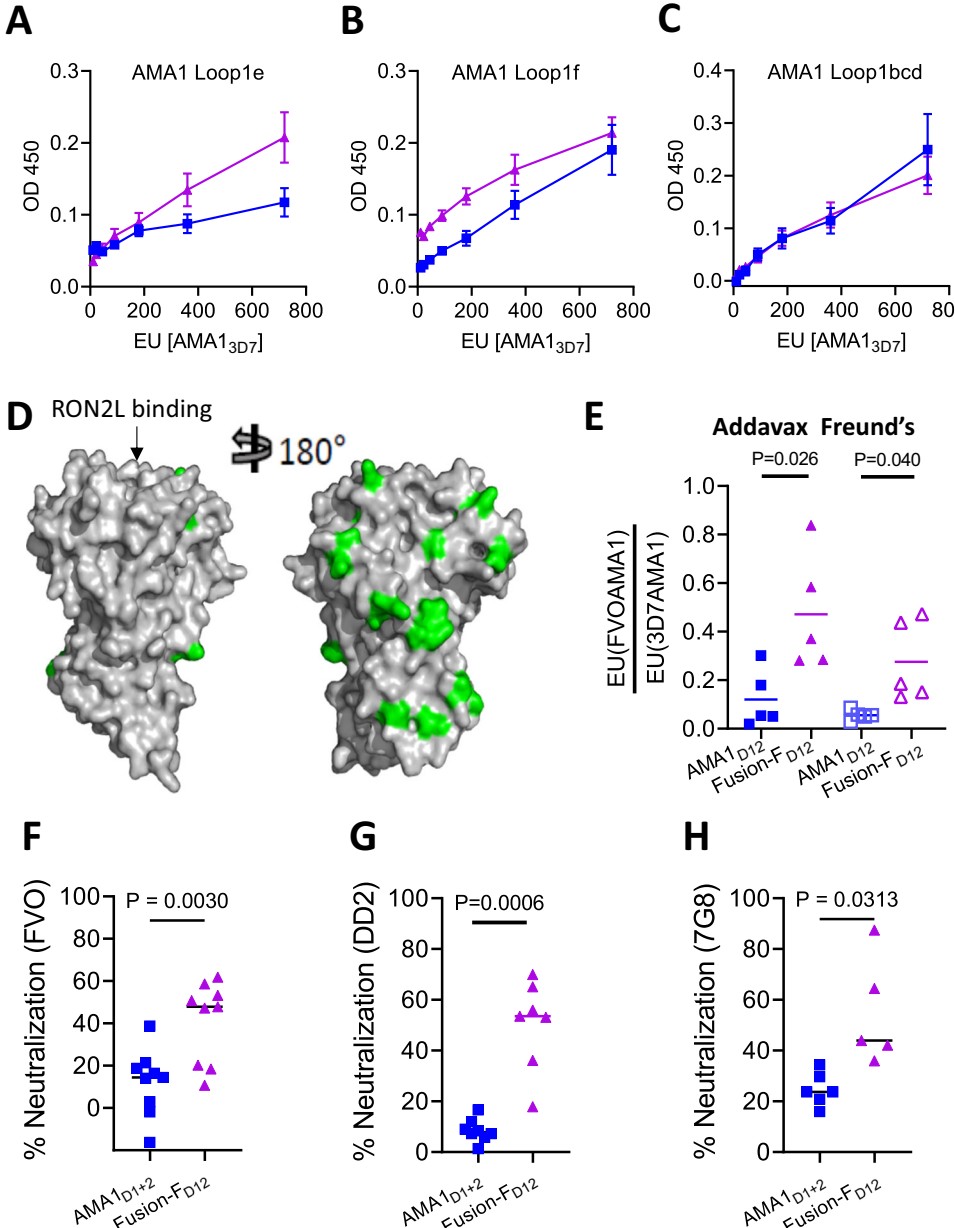

**Fig. 4 | Fusion chimera enhances neutralizing antibodies targeting conserved epitopes on AMA1. A–C** Relative levels of IgG from the Fusion-$F_{D12}$ (purple) and apoAMA1$_{D12}$ (blue) groups targeting AMA1 loop1e (**A**), loop 1 f (**B**) and loops 1bcd (**C**). The *x*-axis indicates the amount of total AMA1-specific antibody titer (EU). Data are mean ± SEM (*n* = 5 per group) and each data point is the average of two replicates. **D** Comparison of polymorphisms in the FVO AMA1 relative to 3D7 AMA1. The conserved and polymorphic face of AMA1 is shown with the polymorphisms marked in green. RON2L binding site on AMA1 is indicated by an arrow. **E** Ratio of IgG binding to non-vaccine type FVOAMA1 vs. vaccine-type 3D7AMA1. Data are shown for individual animals (*n* = 5 per group) and each data point is the average of three replicates. Horizontal line marks the mean for each group. Two-tailed Welch's *t*-test was performed to compare differences between groups. **F** In vitro neutralization assay against PfFVO parasites using purified IgG from AMA1$_{D12}$ and Fusion-$F_{D12}$ groups normalized for anti-3D7AMA1 titer (35,000 EU. Data are shown for individual animals (*n* = 9 per group) and each data point is the average of two

replicates. Horizontal line shows mean neutralizing activity of each group. Two-tailed Welch's *t*-test was performed to compare differences between groups. **G** In vitro neutralization assay against PfDD2 parasites using purified IgG from AMA1$_{D12}$ and Fusion-$F_{D12}$ groups normalized for anti-3D7AMA1 titer (40,000 EU). Data are shown for individual animals (*n* = 7, 8 per group for AMA1$_{D1+2}$ and Fusion-$F_{D12}$ respectively based on sample availability) and each data point is the average of two replicates. Horizontal line shows mean neutralizing activity of each group. Two-tailed Welch's *t*-test was performed to compare differences between groups. **H** In vitro neutralization assay against Pf7G8 parasites using purified IgG from AMA1$_{D12}$ and Fusion-$F_{D12}$ groups normalized for anti-3D7AMA1 titer (60,000 EU). Data are shown for individual animals (*n* = 5, 6 per group for AMA1$_{D1+2}$ and Fusion-$F_{D12}$ respectively based on sample availability) and each data point is the average of two replicates. Horizontal line shows mean neutralizing activity of each group. Two-tailed Welch's *t*-test was performed to compare differences between groups.

## Discussion

Stalling of progress in malaria control in recent years and the threat of a resurgence in *P. falciparum* malaria-related deaths[1] highlight the urgent need for an effective malaria vaccine. Age-related clinical protection is observed in malaria endemic population[28] and infection-

induced antibodies can clear parasites and resolve clinical symptoms in children[29]. Identifying the target(s) of such protective antibodies can help develop vaccines to prevent disease. Candidate antigens have shown promise in preclinical studies, however clinical trials have so far not proven successful[30]. AMA1 is a leading vaccine candidate as

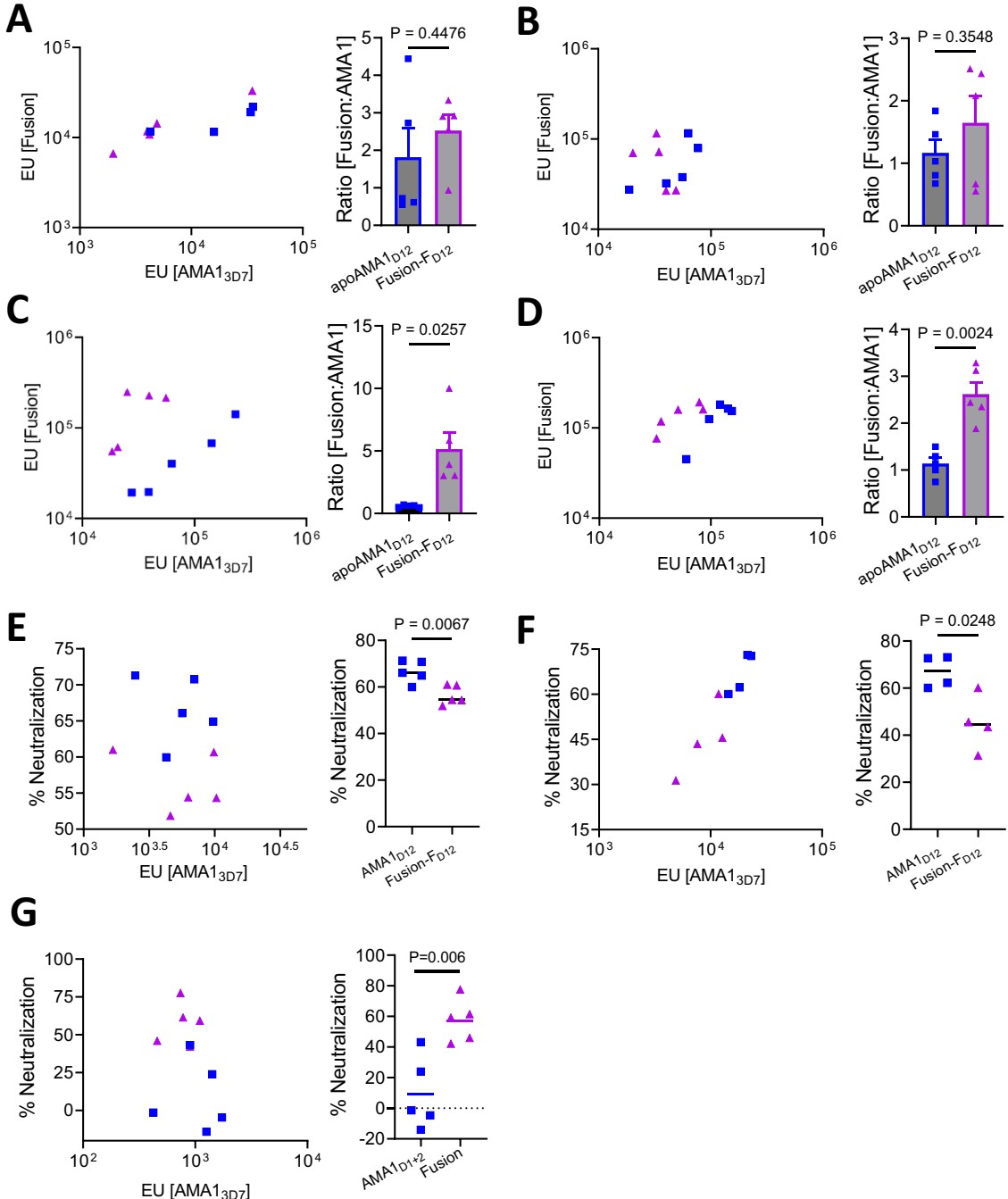

**Fig. 5 | Antibody specificity and neutralizing activity differences in serum and affinity purified IgG. A**, **B** (left panels) Serum IgG titer in animals immunized with the Fusion-$F_{D12}$ (purple triangle) or apoAMA1$_{D12}$ (blue square) antigens in AddaVax (**A**) and Freund's (**B**). The $x$-axis shows antibody titer against apoAMA1$_{3D7}$ and $y$-axis shows antibody titer against the Fusion-$F_{D12}$ antigen. Data are shown for individual animals ($n$ = 5 per group) and each data point is the average of two replicates. (**A**, **B** −right panels) Fusion-$F_{D12}$ to apoAMA1 IgG ratio from data shown to the left in **A** and **B**, respectively. Two-tailed Welch's $t$-test was performed to compare differences between groups. Data are shown as mean ± SEM for individual animals ($n$ = 5 per group) and each data point is the average of two replicates. **C**, **D** (left panels) Antibody titer in 10 mg/mL of purified IgG from animals immunized with the Fusion-$F_{D12}$ (purple triangle) or apoAMA1$_{D12}$ (blue square) antigens in AddaVax (**C**) and Freund's (**D**). The $x$-axis shows antibody titer against apoAMA1$_{3D7}$ and $y$-axis shows antibody titer against the Fusion-$F_{D12}$ antigen. Data are shown for individual animals ($n$ = 5 per group) and each data point is the average of two replicates. (**C**, **D** −right panels) Fusion-$F_{D12}$ to apoAMA1 IgG ratio from data shown to the left in **C** and **D**, respectively. Two-tailed Welch's $t$-test was performed to compare differences between groups. Data are shown as mean ± SEM for individual animals ($n$ = 5 per group) and each data point is the average of two replicates. **E** Left−Relationship between anti-AMA1 antibody titer ($x$-axis) in the purified IgG and neutralizing (2-cycle) activity at 2 mg/mL total IgG ($y$-axis) between Fusion-$F_{D12}$ (purple triangle) and apoAMA1$_{D12}$ (blue square) antigens in AddaVax. Right−Comparison of data shown in left using Two-tailed Welch's $t$-test. **F** Left−Relationship between anti-AMA1 antibody titer ($x$-axis) in the purified IgG and neutralizing (2-cycle) activity at 1.4 mg/mL total IgG ($y$-axis) between Fusion-$F_{D12}$ (purple triangle) and apoAMA1$_{D12}$ (blue square) antigens in Freund's. Right−Comparison of data shown in left using Two-tailed Welch's $t$-test. **G** Left−Relationship between anti-AMA1 antibody titer in serum ($x$-axis) and neutralizing (2-cycle) activity in 2% serum ($y$-axis) between Fusion-$F_{D12}$ (purple triangle) and apoAMA1$_{D12}$ (blue square) antigens in Freund's adjuvant. Right−Comparison of data shown in left using Two-tailed Welch's $t$-test.

antibodies neutralize parasites in vitro and afford partial protection in NHP models of human malaria[17,31,32]. Numerous polymorphisms in AMA1 surrounding the binding site of its ligand RON2 suggest it is an important target of human neutralizing antibodies[18]. However in phase 2 clinical trials, no protection was observed even against vaccine-type parasites despite the vaccine generating robust anti-AMA1 antibody titer[12,15], highlighting the need to improve antibody quality. Previously, we have shown that a binary complex vaccine formed by mixing two parasite proteins, AMA1 and RON2L, a 49 amino acid peptide from its ligand RON2, confers superior protection against *P. falciparum* in NHPs by enhancing the proportion of neutralizing antibodies targeting critical epitopes on AMA1[17]. Similarly, *Toxoplasma gondii* AMA1 + RON2L binary complex vaccine protects against acute and chronic forms of infection[33], demonstrating the vaccine potential of this target in different Apicomplexan parasites.

In this study using structure-guided design, we engineered a chimeric immunogen to mimic the structure of the *P. falciparum* receptor-ligand (AMA1-RON2L) complex. We first determined the minimal region of AMA1 that is sufficient to enhance antibody quality upon binding to RON2L. The structure of one of the fusion designs (Fusion-$F_{D12}$) was determined to 1.55 Å resolution and found to mirror the structure of the binary complex. Immunization studies performed in rats showed adjuvant-dependent changes in antibody quantity and quality. Importantly, the Fusion-$F_{D12}$ chimera, like the binary complex, generated higher proportion of cross-neutralizing antibodies targeting conserved epitopes on AMA1. An interesting observation was the difference in activity of antibodies in the serum and following affinity purification of IgG. While Fusion-$F_{D12}$ immunized rat serum exhibited greater parasite neutralization, affinity purified IgG had lower neutralizing activity compared to apoAMA1$_{D12}$ immunized animals. Such differences in neutralizing activity in the serum and purified IgG may explain the superior protection observed in vivo compared to moderate increases in neutralizing activity observed in vitro using affinity purified IgG following AMA1 + RON2L binary complex immunization[17].

The ability of the Fusion-$F_{D12}$ chimeric antigen to enhance neutralization of both vaccine type and non-vaccine type parasites highlight the potential of using structure-guided antigen design to improve antibody quality against AMA1. However, this was at the cost of reduced immunogenicity against AMA1 and generation of fusion-specific antibodies that may not be functional. Eliminating fusion-specific epitopes through targeted mutagenesis or glycan masking could further promote broadly neutralizing antibody responses against AMA1. Additionally, identifying targets of the broadly neutralizing antibodies generated by the chimera and incorporating AMA1 polymorphisms in the fusion design can broaden the antibody repertoire while improving antibody quality to effectively neutralize all *P. falciparum* parasites. Other parasite antigens like RH5 that are targets of potent neutralizing antibodies also have interaction partners[34,35]. Applying structure-guided antigen design to focus antibody responses against critical epitopes can help develop effective next generation malaria vaccines.

## Methods

### Cloning, expression in Sf9 cells, and protein purification
A clone encoding the full ectodomain (domains 1, 2, and 3) of PfAMA1 (PfAMA1 N25 to K546) sequences from 3D7 and HP41 strains with a section of the DII loop (T358 to K370) replaced by the PfRON2L sequence (T2023 to S2059) and a six residue in the forward (Fusion-$F_{D123}$) or reverse (Fusion-$R_{D123}$) direction with suitable linkers (Suppl Fig. 2) was codon optimized for insect cells and synthesized by GenScript. A second set of constructs encompassing PfAMA1 D1 and D2 (N104 to E438) corresponding to both 3D7 and HP41 AMA1 sequences with the embedded PfRON2L insertion (Fusion-$F_{D12}$ and Fusion-$R_{D12}$) was amplified by PCR and sub-cloned into a modified pAcGP67b vector with a TEV-cleavable N-terminal hexahistidine

tag. Viruses for insect cell protein production were generated and amplified using established protocols[36]. Recombinant proteins including apoAMA1$_{D12}$ was produced and purified from the insect cell culture supernatant as described previously[37]. Briefly, Ni-affinity chromatography was followed by overnight TEV cleavage, and further purification using a Superdex 75 HiLoad column equilibrated in Hepes Buffer Saline (HBS) (20 mM Hepes pH 7.5, 150 mM NaCl). The final yield was approximately 2.0 mg of purified protein per liter of insect cell culture. SDS-PAGE and western blotting analysis of purified proteins were performed conformation-specific mAbs 1F9[11] and 4G2[23]. Pichia pastoris expression of full length ectodomains of 3D7 and FVO AMA1 (residues 25-546) and their purification is described elsewhere[38]. RON2L-Fc fusion was expressed in HEK293T cells and affinity purified on protein A column.

### Crystallization and data collection
Crystals of the Fusion-$F_{D12}$ chimera were originally identified in the ProPlex screen (Molecular Dimensions) using sitting drops at 18 °C. The final, refined drops consisted of 0.8 μL Fusion-$F_{D12}$ at 15 mg/mL with 0.8 μL of reservoir solution (0.1 M Tris-HCl pH 7.0, 1.5 M lithium sulfate) and were equilibrated against 120 μL of reservoir solution. Crystals grew to a final size within five days, were cryoprotected in saturated lithium sulfate, and flash cooled in liquid nitrogen. Diffraction data were collected on beamline 08ID-1 at the Canadian Light Source (CLS).

### Data processing, structure solution, and refinement
Diffraction data were processed to 1.55 Å resolution using Imosflm[39] and Aimless[40] in the CCP4 suite of programs[41]. The structure of Fusion-$F_{D12}$ was solved by molecular replacement in Phaser[42] using a model of PfAMA1 from the co-structure with PfRON2L (PDB ID-3ZWZ), but with the peptide removed to prevent model bias. Tracing of the PfRON2L sequence linker, as well as other model building and selection of solvent atoms, was performed manually in COOT[43]. The model was refined in Phenix.refine[44], and complete structural validation was performed with Molprobity[45], including analysis of the Ramachandran plots, which showed 98% of residues in the most favored conformations. Five percent of reflections were set aside for calculation of $R_{free}$. Data collection and refinement statistics are presented in Suppl Table 1. The atomic coordinates and structure factors for Fusion-$F_{D12}$ chimera have been deposited in the Protein Data Bank under the following code: 8G6B [https://www.rcsb.org/structure/8g6b].

### Peptide synthesis
RON2L and AMA1 loop peptides 1bcd, 1e and 1f were synthesized by LifeTein (South Plainfield, NJ). Quality control included mass spectrometry and high-performance liquid chromatography for assessing purity. The peptides utilized were found to be over 95% pure based on the results obtained from these evaluations.

### Animals, adjuvants, and antigen dose
Outbred female Sprague Dawley rats (Charles River Laboratory) that were 5-6 week old ($n = 4-5$ per group) were used in all our studies. All animal experiments were approved by the Johns Hopkins Animal Care and Use Committee (ACUC), under protocol RA22H291. The AMA1-RON2L complex was prepared immediately before immunization by mixing the two proteins at a 1:3 gram ratio as previously described[8,16,17]. The mixture was then incubated for 30 min at room temperature (RT) to spontaneously form a complex[46]. Antigens diluted in 1x phosphate-buffered saline (PBS, pH 7.4) and mixed with equal volume of AddaVax (Invivogen) or Freund's (Sigma, complete or incomplete) adjuvant. AddaVax adjuvant was mixed by pipetting while antigen in Freund's (complete or incomplete) were emulsified by vortexing for 30 min. Antigen were administered subcutaneously three times at 2-week intervals. Animals receiving antigen in the Freund's group received

their first immunization in complete Freund's adjuvant and the two boosts in incomplete Freund's adjuvant. All data except that shown in Fig. 1 and Suppl Fig. 4 used 15 μg antigen per animal per dose while data in Fig. 1 and Suppl Fig. 4 used 10 μg AMA1$_{(D123 \text{ or } D12)}$ or 10 μg AMA1$_{(D123 \text{ or } D12)}$ + 30 μgRON2L binary complex per animal per dose. Serum aliquots were mixed with 50 μL of human RBCs for 1 h at RT on a rotator to reduce non-specific binding activity in neutralization assays and stored at −80 °C.

## Sample preparation for immunogenicity evaluation and parasite neutralization assays

IgG were affinity purified on pre-equilibrated protein G column (GE health sciences), dialyzed against RPMI 1640 medium, concentrated on AMICON ULTRA 3k cutoff spin columns (Millipore) and sterile filtered using AMICON Ultrafree-MC Sterile 0.22 μm tubes (Millipore). Serum samples were diluted in RPMI 1640 medium and sterile filtered as described above.

## Enzyme-linked immunosorbent assay (ELISA)

The assay was performed as described[47] with some modifications. Immulon 4HBX flat bottom 96-well plates (Thermo Fisher) were coated with indicated recombinant antigens (0.5 μg/mL) and incubated overnight at 4 °C. Antigen-specific ELISA units were first determined by generating a standard curve using serially diluted serum made previously by mixing equal volumes of serum from rats immunized with AMA1 and AMA1 + RON2L binary complex. The dilution that produced an OD$_{405}$ of 1 was identified, and the reciprocal of that dilution was used to assign ELISA units to the standards. All samples were then tested against this same standard. Antibody binding was detected using immunoglobulin-specific (IgG, Cat# 612-1325, Rockland or IgM, Cat # 31476, Thermofisher Scientific) or isotype-specific (IgG1, Cat# PA1-84708, Thermofisher Scientific or IgG2a, Cat# PA1-84709 Thermofisher Scientific) secondary antibody conjugated to HRP (Thermofisher Scientific).

Antibody titer against various AMA1 loops (1bcd, 1e and 1f) and RON2L peptide were measured by coating plates with 2 μg/mL of the respective peptides overnight at 4 °C. To assess the relative proportions of antibodies in the different groups, each sample was normalized to contain the same quantity of anti-AMA1 titer (ELISA units) (Suppl Fig. 6A).

Competition ELISAs were performed as above with some modifications. Briefly, plates were incubated with the coating antigen (0.5 μg/mL) for 2 h at 37 °C. Dilutions of IgG from each group was preincubated with 2 μM r3D7AMA1 for 1 h at RT before adding to ELISA plate. The proportion of antibodies binding to the target antigen was measured relative to the no competition wells.

## Parasite maintenance

3D7 and FVO strain parasites were maintained in vitro as described previously[48]. Parasites were cultured in RPMI 1640 with 25 mM Hepes, 50 μg/mL hypoxanthine (KD Medical), 0.23% sodium bicarbonate (Gibco), and 0.5% Albumax (Invitrogen) using human erythrocytes (Johns Hopkins IRB: NA_00019050) at 2% hematocrit and incubated at 37 °C. Cultures were monitored daily using Giemsa-stained thin blood smears.

## Parasite neutralization assay using affinity purified IgG

1-cycle assay (NIH): Infected RBCs were incubated with affinity purified IgG (2 mg/mL) for 40 h. Parasitemia was quantified by biochemical measurement using a Pf lactate dehydrogenase assay as described previously[49].

2-cycle assay (JHU): The assay was setup as described above with indicated amounts of total IgG, incubated for 72 h at 37 °C and stained with SYBR green dye (Invitrogen). Infected cells were counted using a AttuneNxT flowcytometer connected to an autosampler. All assays were carried out in duplicate. Percent inhibition of invasion=1-(% parasitemia in test well/% parasitemia in medium control well). Strong correlation between 1-cycle and 2-cycle neutralization assay was observed (Suppl Fig. 6B).

## Parasite neutralization assay using rat serum

Heat inactivated individual serum (up to 2% in the assay wells) were used in the 2-cycle neutralization assay as described above. Non immunized rat serum at the same dilutions were used as controls. Percent inhibition of invasion = 1-(% parasitemia in test serum well/% parasitemia in control serum well).

## Statistical analysis

All statistical computations were performed with Graphpad Prism software v9.5.1 using Two-tailed $t$-test with Welch's correction, one-way ANOVA with Brown-Forsythe and Welch's correction where indicated. $P$-values are shown in the figures and additional details including group size are provided within each figure legend.

## Reporting summary

Further information on research design is available in the Nature Portfolio Reporting Summary linked to this article.

## Data availability

All data supporting the findings of this study are available within the paper and it's Supplementary Information. All raw data and analysis are provided as a Source Data file. Crystal structure atomic coordinates and structure factors have been deposited in the Protein Data Bank with PDB ID 8G6B. Source data are provided with this paper.

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

## Acknowledgements

We thank Dr. Louis H Miller, NIH, for valuable discussions during the study. We also thank the Srinivasan lab members for helpful discussions. S.Y. was funded by the NIH (T32 AI138953). D.S was supported by Johns Hopkins Malaria Research Institute Postdoctoral fellowship. The GIA and analysis conducted at National Institute of Allergy and Infectious Disease (NIAID) were supported by the intramural program of the NIAID/NIH. The GIA activity was also supported by an Interagency Agreement (AID-GH-T-15-00001) with the U.S. Agency for International Development (USAID) Malaria Vaccine Development Program. The findings and conclusions in

this report are those of the authors and do not necessarily represent the official position of USAID. M.J.B was funded by Canadian Institutes of Health Research (CIHR 148596) and P.S. was in part funded by the NIH (R01AI155598). This work was supported by the Johns Hopkins Malaria Research Institute and the Bloomberg Philanthropies.

## Author contributions

P.S. Conceived the study. P.S., M.J.B., C.A.L., K.M., S.Y designed the experiments.. S.Y., V.V., and P.S performed animal experiments. M.L.P., R.R., and M.J.B. performed structural biology experiments. M.J.B. built and refined structural models. S.Y., V.V., A.D., D.S. performed parasite culture, ELISA and neutralization assays. All authors analyzed the results and contributed to figure preparation and the reviewing and editing of the manuscript. P.S., S.Y., and M.J.B. wrote the manuscript.

## Competing interests

P.S. is a named inventor on patents related to this work. The remaining authors declare no competing interests.
