## [Peer Review File · Nature Communications]

Structure guided mimicry of an essential *P. falciparum* receptor-ligand complex enhances cross neutralizing antibodiesEditorial Note: Parts of this Peer Review File have been redacted as indicated to maintain the confidentiality of unpublished data.

Reviewers' Comments:

Reviewer #1:

Remarks to the Author:

In this work, Dr Srinivasan and co-authors report the development of a chimeric antigen that is aimed to mimic the structure of the AMA1-RON2 receptor-ligand complex. The rationale for this approach is to produce a single immunogen amenable to use as a vaccine antigen, that could replicate the increased proportion of neutralising antibodies observed on immunisation with the AMA1-RON2L receptor-ligand complex.

The authors have described the rationale and process of design and production that eventually led to the development of the Fusion-FD12 chimeric antigen. This study has made a number of findings that will be of interest to the field, however I feel that it is the nature of the overall approach used that is the most significant and interesting contribution that this study makes.

Selected highlights of this study are:

1. AMA1 domains 1&2 were sufficient to replicate the enhancement effect previously observed by the AMA1 + RON2L ligand complex. This is advantageous since this study reports that fusion constructs containing AMA1 domains 1-3 were unsuccessful.
2. This study produced a fusion antigen, FusionFD12, which is the main focus of most later components of the study. Structural data show that Fusion-FD12 is a close structural mimic of the binary receptor-ligand complex, but there are some notable structural differences.
3. Purified IgG from animals immunised with Fusion-FD12 had lower AMA-1 specific antibody titre than those immunised with AMA1D12, in two adjuvant systems (fig 3A).
4. Purified IgG from animals immunised with Fusion-FD12 had lower parasite-neutralisation activity, even after adjustment for AMA1-specific titre (fig 3B).
5. Animals immunised with FusionFD12 made antibodies that were specific to the fusion protein that did not react with AMA-1. An interesting additional observation in the competition ELISA was that removal of AMA-1 specific antibodies from IgG from FusionFD12 immunised animals increased the reactivity to the fusion.
6. Animals immunised with FusionFD12 had a higher proportion of total AMA-1-reactive antibodies that react against two conserved loops (1e and 1f) - figure 4A-4C.

In my view, all of the claims above are well supported by the data presented. I have some concerns and uncertainties regarding the extent to which the claims noted below are supported in the manuscript.

A. Immunisation with Fusion-FD12 neutralised parasites more 'efficiently' than apoAMA1 immune sera, despite lower anti-AMA-1 titre. (L28-30). I think that the key word here is 'efficiently' and may have missed the data that directly support this.

B. Immunisation with Fusion-FD12 targeted conserved epitopes, resulting in greater neutralisation of parasites carrying non-homologous AMA-1. (L31-30, L222-225). Figure 4F shows this data, with a p value of 0.058: this claim is therefore not supported within a commonly used threshold of $p < 0.05$ (null hypothesis not rejected).

C. The Fusion-FD12 chimera... generated higher levels of cross neutralising antibodies targeting conserved epitopes. (L287-289). What is stated in results is that the proportion of AMA-1 specific IgG

that reacts with loops 1e and loop 1f is higher in fusion sera. This is not the same as 'higher levels', which would require an higher absolute titre.

D. The vaccine platform "can be enhanced by incorporating polymorphisms in AMA1 to effectively neutralise all *P. falciparum* parasites" (L34-36). I do not see evidence in this manuscript that directly supports this claim.

Overall, the approach taken by the authors is novel and exciting, and this represents a very substantial body of work that has been carried out to a high standard and will be of interest to the field. I feel that the presentation of the results, in particular those highlighted in the abstract, emphasises some secondary interpretations of the data that are based on ratios of normalised ELISA reactivity at the cost of absolute values. In protection against a parasite, it is absolute levels of neutralisation that are important, and as figure 3B demonstrates, this is lower with the Fusion FD12. This key finding is omitted from the abstract in favour of the more positive claims and I think that the wording of these claims could be modified to reflect the results more realistically (i.e. including those findings where the comparison to the AMA1-RON2L complex is less favourable), since these aspects are also important in the reporting of this work.

Regarding the isotype work mentioned at the end of the results, it is interesting to note that the FusionFD12 antigen raised higher levels of IgG1 compared with IgG2a. It is well established that Protein G has poor binding of Rat IgG1 and good binding of Rat IgG2a (<https://uk.neb.com/tools-and-resources/selection-charts/affinity-of-protein-ag-for-igg-types-from-different-species>), and I wonder whether the authors have considered this as a possible explanation for the differential results obtained between serum and purified IgG results. Protein L would be another interesting possibility (though it would only purify antibodies with kappa chains).

I have no concerns regarding the methods section - this is well written and would allow replication of the wear lab experiments. I cannot see any reporting of the statistics software approaches that were used, other than in figure legends.

I have made some additional notes regarding areas of the manuscript that I hope will be useful to the authors:

L24: The work cited (10) was performed in Aotus monkeys, and reference 9 is in rodent. Given the clinical context of the preceding sentence, it would be useful to add text highlighting the distinction, to avoid misleading.

L42 "...it is the parasites growing within the safety of the host cell that causes clinical symptoms" - it is during this phase of the infection that clinical symptoms become apparent, but those who are clinically immune (non-sterilising) do not show symptoms despite blood-stage infection. The phrasing should be changed to make more clear the point being made here, or supported by citation.

Line 158 - SuppFig3 - it is claimed that antibody titres resulting from immunisations in Freund's adjuvant have higher titres, but unclear whether statistical analysis has been performed?

Figure 1 legend: "Data are presented for individual animals (n=4) done in triplicate ". It is unclear from the legend whether this refers to experimental replication or technical replication (multiple wells) - most likely the latter. I think that changing this to state that there are three technical replicates of each antibody sample would be clearer.

L160 - the text states that purified IgG from FusionFD12 immunised animals had lower AMA-1 specific titres than apoAMA1D12-immunized animals in both adjuvants, citing figure 3A. Figure 3A would only appear to support for the Freund's group based on P values. I'd suggest these should be checked since the difference does look substantial in both adjuvants (may be due to high variance in 1 group?).

Line 191 - this refers to Supp Figure 3C - this is not presented as that figure has only panels A and B.

Line 220 - reference to Supp Fig 4D - this display of data is not easy to interpret the intermediate level as indicated in the text. A swarm plot or strip plot would facilitate this comparison. 3D7AMA1+RON2L does not have a designated colour (I believe it should be green).

SuppFig 2C: A bar chart is shown here, however it is unclear what the whiskers indicate (SEM, SD, range), or what form of replication was performed. This should be noted in the legend.

Fig 2D - typo (asterisks)

Fig 4A-C - the figure is titled as 'Proportion of IgG...' however it does not directly indicate the proportion of IgG, for which values would fall between 0 and 1. These figures indicate the relative reactivity against the loop antigens for the same AMA1 titre.

Line 233 - 236 - these lines claim increased fusion-specific titres in fusionFD12 immunised group in purified IgG, not observed in serum, citing figures 5A, B, D & E. This can be read from the graphs but it is not a direct way to display the data for this interpretation. It is also not clear whether statistical analyses have been performed to back up the claim?

Figure Colour: Purple and blue are extensively used to represent Fusion-FD12 and apoAMA1d12 antigens throughout the figures. In figure 4A-C there is no difference in the shape of the points that would allow those with very common colour vision deficiencies ("red/green color blindness") or those printing the manuscript in grayscale to easily distinguish series.

Reviewer #2:

Remarks to the Author:

The manuscript by Yanik et al., entitled "Structure guided mimicry of an essential *P. falciparum* receptor-ligand complex enhances cross neutralizing antibodies" designed and characterized a structure based chimeric immunogen of the AMA1-RON2L complex. The authors determined the high-resolution crystal structure of the chimeric immunogen and showed that it looked similar to a previously determined crystal structure of the AMA1-RON2L complex. They then performed immunization studies in rats using different adjuvants and immunogens and showed that immune sera (but not purified IgG) from rats immunized with Fusion-FD12 neutralized parasites more efficiently than sera from rats immunized with apoAMA1 alone. Furthermore, they showed that anti Fusion-FD12 antibodies targeted conserved epitopes on AMA1 and were better at neutralizing FVO parasites.

Collectively, along with the Patel manuscript, these are well-designed and important studies for the malaria vaccine field, utilizing structural biology to develop novel immunogens, and should be of interest to the community. The crystal structure is well described and of excellent quality. The immunization studies, whilst hard to follow at times, were very interesting.

Major Comments

The difference in neutralization that the authors observed between sera and IgG from rats immunized with apoAMA1 and Fusion-FD12 is fascinating and warrants further investigation.

- 1) Is the difference also seen when using Addavax as the adjuvant or is it just when using Freund's?
- 2) Since protein G purification should purify all rat IgG subtypes reasonably well and IgA was not observed, does the greater proportion of IgM in the Fusion-FD12 serum compared to apoAMA1 explain this difference? Does IgM specific to AMA1 correlate with neutralization?

Line 256-258 - The authors state that since only low levels of RON2L directed antibodies were

detected in the Fusion-FD12 group, that they are not likely to influence neutralizing activity. However it's not clear, at least to me, that the levels are low. They appear to have comparable OD450 values to AMA1 Loop1 peptides (Figures 4A-C). Perhaps they were tested at a higher concentration? In any event, in the reference cited, polyclonal anti-RON2L antibodies inhibited both 3D7 and FVO parasites well (~80-90% inhibition) at 2mg/mL, providing another explanation for cross-species neutralization by Fusion-FD12. Perhaps the authors could try depleting anti-RON2L antibodies from sera or IgG from rats immunized with Fusion-FD12 and then testing for neutralization, so see if inhibition and cross-species neutralization is comparable or weaker?

Minor Comments

In your abstract you state 'immunization with Fusion-FD12 enhanced antibodies targeting conserved epitopes on AMA1 resulting in greater neutralization of non-vaccine type parasites.' I suggest you use a different word than 'resulting'. The presence of antibodies targeting conserved epitopes only correlates with neutralization of non-vaccine parasites.

Line 88 – Perhaps change “and some AMA1 antibodies target domain 3” to “and some Inhibitory antibodies target domain 3”?

Line 106 - Fusion-RD12 isn't in Suppl figure 1B or 2A and 2B and should be included if mentioned.

Line 130 – In figure 2D Box 3, how did you calculate the difference in the Beta-strand lengths? Is the main chain hydrogen-bonding pattern similar to the AMA-RON2L crystal structure? How did you calculate secondary structure? And how did you measure the 1.8 Å difference? The reason I ask is that they look very similar in the figure and Pymol doesn't always perfectly predict secondary structure.

Line 139 – N136 or E136?

Line 141 - Do residues in loop 1e form crystal contacts in this structure or in 3zwz? Does that explain the difference in this loop? Are any of the differences due to differences in sequence between 3D7 and HP41?

Line 157 – group f?

Suppl Figure 3B – Is there a data point missing in the AMA1D12 vs 3D7AMA1 group?

Line 181 – Are any of these differences in electrostatics due to the difference in sequence between 3D7 and HP41? For example, in loop 1bcd, there is an E to K polymorphism (Suppl Fig. 1).

Line 189 - How did you calculate the ratio between EU(Fusion-FD12) vs EU(3D7AMA1) in Fig. 3D? It's unclear to me how it's possible to get a negative value.

Line 253 – Suppl Fig. 6C shows a greater proportion of IgM type antibodies against apoAMA1, not the fusion chimera, in the Fusion-FD12 immunized serum compared to apoAMA1D12. Is this correct or just mislabelled?

Suppl Fig. 2E – Change Q35 to Q359

Response to reviewers

We thank the reviewers for their positive comments and valuable suggestions that helped improve our manuscript.

We have now tested IgG from a second independent study comparing antibodies induced by the Fusion-F_{D12} vs apoAMA1_{D12} against vaccine-heterologous parasite PFFVO. Furthermore, we evaluated neutralizing activity of these antibodies against two other heterologous parasite (PfDD2 and Pf7G8 strains). These data provide additional evidence for the enhancement in cross-neutralizing antibodies generated by Fusion-F_{D12} antigen compared to apoAMA1_{D12}.

We have also clarified the statistical analysis performed in each figure. Statistical analysis were always performed on biological replicate not technical replicates. We have clarified this point further under each figure legend (eg., *Data are presented for individual animals (n=4 per group) and each data point is the average of three replicates*).

These studies can pave the way for a better understanding of the cross-neutralizing antibodies targeting AMA1 and help develop next generation AMA1-targeting vaccines to prevent both infection and disease. Following these additional new data showing significant enhancement in cross neutralization of genetically distinct *P.falciparum* strains by antibodies generated by the structure-guided fusion antigen and the clarification of statistical analysis as well as other reviewer suggestions, we hope the manuscript is now suitable for publication.

Please see below a point-by-point response to the reviewers comments.

Reviewer #1 (Remarks to the Author):

In this work, Dr Srinivasan and co-authors report the development of a chimeric antigen that is aimed to mimic the structure of the AMA1-RON2 receptor-ligand complex. The rationale for this approach is to produce a single immunogen amenable to use as a vaccine antigen that could replicate the increased proportion of neutralising antibodies observed on immunisation with the AMA1-RON2L receptor-ligand complex.

The authors have described the rationale and process of design and production that eventually led to the development of the Fusion-FD12 chimeric antigen. This study has made a number of findings that will be of interest to the field, however I feel that it is the nature of the overall approach used that is the most significant and interesting contribution that this study makes.

Selected highlights of this study are:

1. AMA1 domains 1&2 were sufficient to replicate the enhancement effect previously observed by the AMA1 + RON2L ligand complex. This is advantageous since this study reports that fusion constructs containing AMA1 domains 1-3 were unsuccessful.
2. This study produced a fusion antigen, FusionFD12, which is the main focus of most later

components of the study. Structural data show that Fusion-FD12 is a close structural mimic of the binary receptor-ligand complex, but there are some notable structural differences.

3. Purified IgG from animals immunised with Fusion-FD12 had lower AMA-1 specific antibody titre than those immunised with AMA1D12, in two adjuvant systems (fig 3A).

4. Purified IgG from animals immunised with Fusion-FD12 had lower parasite-neutralisation activity, even after adjustment for AMA1-specific titre (fig 3B).

5. Animals immunised with FusionFD12 made antibodies that were specific to the fusion protein that did not react with AMA-1. An interesting additional observation in the competition ELISA was that removal of AMA-1 specific antibodies from IgG from FusionFD12 immunised animals increased the reactivity to the fusion.

6. Animals immunised with FusionFD12 had a higher proportion of total AMA-1-reactive antibodies that react against two conserved loops (1e and 1f) - figure 4A-4C.

In my view, all of the claims above are well supported by the data presented. I have some concerns and uncertainties regarding the extent to which the claims noted below are supported in the manuscript.

We thank the reviewer for the encouraging comments. We are grateful for the appreciation of the novelty of the approach used, and the rigor and reproducibility of the presented data. Your insightful suggestions helped improve our manuscript substantially.

A. Immunisation with Fusion-FD12 neutralised parasites more 'efficiently' than apoAMA1 immune sera, despite lower anti-AMA-1 titre. (L28-30). I think that the key word here is 'efficiently' and may have missed the data that directly support this.

This statement is based on data shown in Fig.5G and Suppl Fig. 5A and B that parasite neutralizing activity is higher in the serum of Fusion-F_{DI2} serum compared to apoAMA1_{D12} at similar anti-AMA1 antibody titers.

B. Immunisation with Fusion-FD12 targeted conserved epitopes, resulting in greater neutralisation of parasites carrying non-homologous AMA-1. (L31-30, L222-225). Figure 4F shows this data, with a p value of 0.058: this claim is therefore not supported within a commonly used threshold of $p < 0.05$ (null hypothesis not rejected).

We thank the reviewer for pointing this. The P value approaching statistical significance in the earlier version of this figure (P=0.058) was likely due to sample size (N=5 rats/ per group). We have now performed additional neutralization assays using IgG from Freund's Expt 1 to increase the sample size for each group (N=9 per group). The combined results from the two studies show significantly (P=0.003) higher neutralization of FVO strain by IgG from Fusion-F_{DI2} group compared to apoAMA1_{D12} (Fig 4F).

Furthermore, we also tested the antibodies against two additional heterologous parasite strains DD2 (N=7-8 per group based on sample availability) and 7G8 (N=5-6 per group based on sample availability) (Fig. 4G and 4H).

Our data shows that at the same anti-AMA1 titer, IgG from the Fusion-F_{D12} group neutralized heterologous parasites significantly greater compared to IgG from apoAMA1_{D12}. Together these data strongly suggest that our structure-guided antigen design enhanced the generation of broadly neutralizing antibodies. These results will have significant implications for vaccine development. These are now included in the revised manuscript.

C. The Fusion-FD12 chimera... generated higher levels of cross neutralising antibodies targeting conserved epitopes. (L287-289). What is stated in results is that the proportion of AMA-1 specific IgG that reacts with loops 1e and loop 1f is higher in fusion sera. This is not the same as 'higher levels', which would require an higher absolute titre.

Thank you, we agree. We have changed "levels" to "proportion".

D. The vaccine platform "can be enhanced by incorporating polymorphisms in AMA1 to effectively neutralise all P falciparum parasites" (L34-36). I do not see evidence in this manuscript that directly supports this claim.

We thank the reviewer for this insightful comment. Earlier studies (Ref 21 and PMID: 23429537) have shown that vaccine incorporating multiple AMA1 variants increases the breadth of neutralizing antibodies. Our data showing that AMA1-RON2L binary complex enhances the proportion of neutralizing antibodies (Fig. 1 and Refs 16 and 17) together with the Fusion-F_{D12} protein enhancing cross-neutralizing antibodies (Fig. 4F-H) leads us to believe that incorporating AMA1 polymorphisms into our Fusion design can increase the proportion of neutralizing antibodies targeting both conserved and polymorphic epitopes, i.e., better quality antibodies. We recently developed a vaccine comprising a polyvalent mixture of 6 AMA1 proteins in complex with the single conserved

[Redacted] Fig: Significantly greater neutralization of heterologous lab-adapted strains and field isolates by IgG from rabbits immunized with PV6AMA1-RON2L complex compared to PV6 AMA1-alone (n=6/grp).

RON2L peptide. Interestingly, antibodies induced by this vaccine neutralized several

lab-adapted and filed isolates significantly higher than a mixture of AMA1-alone. Data is shown below only for the reviewers as this is part of a separate manuscript in preparation.

We have now removed the sentence “can be enhanced by incorporating polymorphisms in AMA1 to effectively neutralise all P falciparum parasites” so as to not detract from the important findings of this manuscript.

Overall, the approach taken by the authors is novel and exciting, and this represents a very substantial body of work that has been carried out to a high standard and will be of interest to the field. I feel that the presentation of the results, in particular those highlighted in the abstract, emphasises some secondary interpretations of the data that are based on ratios of normalised ELISA reactivity at the cost of absolute values. In protection against a parasite, it is absolute levels of neutralisation that are important, and as figure 3B demonstrates, this is lower with the Fusion FD12. This key finding is omitted from the abstract in favour of the more positive claims and I think that the wording of these claims could be modified to reflect the results more realistically (i.e. including those findings where the comparison to the AMA1-RON2L complex is less favourable), since these aspects are also important in the reporting of this work.

Thank you, we agree and are excited by the potential for such structure-guided approaches to develop next generation malaria vaccines. We have now amended the abstract to read “Immunization studies showed that Fusion-FD12 immune sera but not purified IgG neutralized vaccine-type parasites more efficiently than apoAMA1 immune sera despite having an overall lower anti-AMA1 titer. Antibodies following protein G purification from the Fusion-FD12 group but not serum showed a bias towards greater fusion-specific antibodies compared to apoAMA1. This suggests an improvement in serum antibody quality against vaccine-type parasite that was lost following IgG purification.” This now should correctly highlight the observed difference between serum and IgG against vaccine-type parasites.

Regarding the isotype work mentioned at the end of the results, it is interesting to note that the FusionFD12 antigen raised higher levels of IgG1 compared with IgG2a. It is well established that Protein G has poor binding of Rat IgG1 and good binding of Rat IgG2a (<https://uk.neb.com/tools-and-resources/selection-charts/affinity-of-protein-ag-for-igg-types-from-different-species>), and I wonder whether the authors have considered this as a possible explanation for the differential results obtained between serum and purified IgG results. Protein L would be another interesting possibility (though it would only purify antibodies with kappa chains).

We thank the reviewer for the insight. We agree that differences in the binding affinity of rat IgG isotypes to Protein G and it would be interesting to see why the FusionFD12 protein may induce specific isotype not induced by apoAMA1. We attempted to purify rat IgG using protein L but the yield was suboptimal and did not allow for further testing.

I have no concerns regarding the methods section - this is well written and would allow replication of the wear lab experiments. I cannot see any reporting of the statistics software approaches that were used, other than in figure legends.

We thank the reviewer for pointing this. While we had included all the statistics performed along with the corresponding figure legends, we have now added a separate section under methods titled “Statistical analysis”.

L24: The work cited (10) was performed in Aotus monkeys, and reference 9 is in rodent. Given the clinical context of the preceding sentence, it would be useful to add text highlighting the distinction, to avoid misleading.

Thank you, we have made this change by adding “...P. yoelli in a mouse model ¹⁶ and P. falciparum in a non-human primate malaria model ¹⁷.”

L42 “...it is the parasites growing within the safety of the host cell that causes clinical symptoms” - it is during this phase of the infection that clinical symptoms become apparent, but those who are clinically immune (non-sterilising) do not show symptoms despite blood-stage infection. The phrasing should be changed to make more clear the point being made here, or supported by citation.

Thank you. We have clarified it by adding “in susceptible individuals”.

Line 158 - SuppFig3 - it is claimed that antibody titres resulting from immunisations in Freund’s adjuvant have higher titres, but unclear whether statistical analysis has been performed? Figure 1 legend: “Data are presented for individual animals (n=4) done in triplicate “. It is unclear from the legend whether this refers to experimental replication or technical replication (multiple wells) - most likely the latter. I think that changing this to state that there are three technical replicates of each antibody sample would be clearer.

We have now clarified this in the figure legend to state “each data point represents the average of three replicates.”

L160 - the text states that purified IgG from FusionFD12 immunised animals had lower AMA-1 specific titres than apoAMA1D12-immunized animals in both adjuvants, citing figure 3A. Figure 3A would only appear to support for the Freund’s group based on P values. I’d suggest these should be checked since the difference does look substantial in both adjuvants (may be due to high variance in 1 group?).

We agree that the differences look substantial in both. However, they are statistically significant in the Freund’s adjuvant study but not AddaVax as shown in Fig. 3A. This may be due to the variance in the AddaVax study as the reviewer pointed. Importantly, the mean anti-3D7 AMA1 titer in the Fusion group is consistently lower compared to AMA1_{D12} in three independent studies using two different adjuvants (Fig. 3 and Suppl Fig.3).

Line 191 - this refers to Supp Figure 3C - this is not present as that figure has only panels A and B.

Thank you, this was not correct. We removed that sentence.

Line 220 - reference to Supp Fig 4D - this display of data is not easy to interpret the intermediate level as indicated in the text. A swarm plot or strip plot would facilitate this comparison. 3D7AMA1+RON2L does not have a designated colour (I believe it should be green).

Thank you we agree. We have now performed statistical analysis comparing the ratio of Ig against FVO (conserved antibodies) to 3D7 (total antibodies) between the three groups (new Suppl Fig. 4C).

Significant difference between apoAMA1_{D12} and Fusion-F_{D12} but not between apoAMA1_{D12} and AMA1_{D12}+RON2L binary complex group was observed. We have included the following sentence to accurately reflect this data “Interestingly, the proportion of IgG binding to FVO AMA1 (conserved epitopes) was intermediate to that of apoAMA1_{D12} and Fusion-F_{D12} groups thought not statistically significant (Suppl Fig. 4C).”

SuppFig 2C: A bar chart is shown here, however it is unclear what the whiskers indicate (SEM, SD, range), or what form of replication was performed. This should be noted in the legend.

We have added the details to the figure legend (new Suppl Fig.2E) as suggested.

Fig 2D - typo (asterisks)

Done

Fig 4A-C - the figure is titled as ‘Proportion of IgG...’ however it does not directly indicate the proportion of IgG, for which values would fall between 0 and 1. These figures indicate the relative reactivity against the loop antigens for the same AMA1 titre.

Thank you, we have changed “Proportion of IgG” to “Relative levels of IgG” to correctly reflect the assay.

Line 233 - 236 - these lines claim increased fusion-specific titres in fusionFD12 immunised group in purified IgG, not observed in serum, citing figures 5A, B, D & E. This can be read from the graphs but it is not a direct way to display the data for this interpretation. It is also not clear whether statistical analyses have been performed to back up the claim?

We have now included in Figure 5A-D analysis of Fusion:apoAMA1 antibody titer ratio. These data show statistically significant difference in the Fusion:apoAMA1 titer ratio in the serum and IgG following ProteinG purification.

Figure Colour: Purple and blue are extensively used to represent Fusion-FD12 and apoAMA1_{D12} antigens throughout the figures. In figure 4A-C there is no difference in the shape of the points that would allow those with very common colour vision deficiencies (“red/green color blindness”) or those printing the manuscript in grayscale to easily distinguish series.

We thank the reviewer for pointing this out. We have now changed the shapes of the individual data points to distinguish between the groups.

Reviewer #2 (Remarks to the Author):

The manuscript by Yanik et al., entitled “Structure guided mimicry of an essential *P. falciparum* receptor-ligand complex enhances cross neutralizing antibodies” designed and characterized a structure based chimeric immunogen of the AMA1-RON2L complex. The authors determined the high-resolution crystal structure of the chimeric immunogen and showed that it looked similar to a previously determined crystal structure of the AMA1-RON2L complex. They then performed immunization studies in rats using different adjuvants and immunogens and showed that immune sera (but not purified IgG) from rats immunized with Fusion-FD12 neutralized parasites more efficiently than sera from rats immunized with apoAMA1 alone. Furthermore, they showed that anti Fusion-FD12 antibodies targeted conserved epitopes on AMA1 and were better at neutralizing FVO parasites.

Collectively, along with the Patel manuscript, these are well-designed and important studies for the malaria vaccine field, utilizing structural biology to develop novel immunogens, and should be of interest to the community. The crystal structure is well described and of excellent quality. The immunization studies, whilst hard to follow at times, were very insightful.

Thank you for the positive feedback and the valuable suggestions that have helped improve the manuscript. Please see below a point-by-point response to the comments.

Major Comments

The difference in neutralization that the authors observed between sera and IgG from rats immunized with apoAMA1 and Fusion-FD12 is fascinating and warrants further investigation.

1) Is the difference also seen when using Addavax as the adjuvant or is it just when using Friends?

Thank you. We agree this is an interesting observation. Our data shows that the proportion of antibodies binding Fusion-F_{D12} vs apoAMA1 is significantly different in the Fusion-FD12 immunized animals following protein G purification in both the AddaVax and Friends study (Fig. 5). Unfortunately, we were unable to test the AddaVax serum in the neutralization assays since all the sample were used for IgG purification. When we noticed the apparent difference in IgG titer against Fusion-FD12 in the Fusion antigen immunized animals, it prompted us to evaluate possible differences in antibody activity in the serum and purified IgG from the subsequent study using Friends adjuvant (Expt 3).

2) Since protein G purification should purify all rat IgG subtypes reasonably well and IgA was not observed, does the greater proportion of IgM in the Fusion-FD12 serum compared to apoAMA1 explain this difference? Does IgM specific to AMA1 correlate with neutralization?

This is a very interesting question. A recent study showed that antigen-specific human mAb of the IgM type isolated from malaria-experienced individuals have potent neutralizing activity. We have not been able to purify sufficient amounts of IgM antibody required for neutralization assays from the relatively small volumes of rat serum. Future studies using large animals like rabbits can help answer that question.

A Spearman correlation analysis indicated a positive correlation between IgM titer and neutralizing activity of the antibodies in the serum from the AMA1 group that was not statistically significant (R: 0.3, P:0.68), likely due to the small sample size (n=5). The role of vaccine induced IgM antibodies is nevertheless an important and interesting question that will be addressed outside of the scope of this manuscript.

Line 256-258 – The authors state that since only low levels of RON2L directed antibodies were detected in the Fusion-FD12 group, that they are not likely to influence neutralizing activity. However it's not clear, at least to me, that the levels are low. They appear to have comparable OD450 values to AMA1 Loop1 peptides (Figures 4A-C). Perhaps they were tested at a higher concentration? In any event, in the reference cited, polyclonal anti-RON2L antibodies inhibited both 3D7 and FVO parasites well (~80-90% inhibition) at 2mg/mL, providing another explanation for cross-species neutralization by Fusion-FD12. Perhaps the authors could try depleting anti-RON2L antibodies from sera or IgG from rats immunized with Fusion-FD12 and then testing for neutralization, so see if inhibition and cross-species neutralization is comparable or weaker?

The OD for RON2L was measured using 1:50 dilution of immunized serum while purified IgG normalized for AMA1 titer was used for AMA1 loop peptides and therefore cannot be directly compared. Importantly, at the 1:50 dilution tested, the OD values between AMA1_{D12} group were not significantly different from Fusion-F_{D12} group (P=0.17). Since AMAD₁₂ group did not have any RON2L, it suggests that these are only mouse serum background.

The study (Ref 8) showing neutralization at 2mg/mL antibody used affinity purified RON2L-specific antibody. Subsequent studies (Ref 16) showed that total IgG (protein G purified) from animals immunized with KLH conjugated PfRON2L peptide had no significant neutralizing activity against P.falciparum. Furthermore, we previously shown animals vaccinated with PyRON2L-KLH did not show growth reduction following P.yoelli challenge (Ref 16). Therefore the poor immunogenicity to RON2L in the absence of KLH conjugation and the negligible anti-RON2L antibodies in comparison to anti-AMA1 antibodies are not likely to contribute to the observed neutralizing activity.

We also attempted to deplete anti-RON2L antibodies using RON2L peptide as the reviewers suggested. However, it was not possible to sufficiently remove all the peptide (4.5kD) following depletion resulting in high background inhibition even no antibody controls. This is due to the small amount of leftover RON2L peptide that by itself blocking parasite invasion by competing for binding to AMA1 on the parasite surface (Ref 8).

Minor Comments

In your abstract you state 'immunization with Fusion-FD12 enhanced antibodies targeting conserved epitopes on AMA1 resulting in greater neutralization of non-vaccine type parasites.' I suggest you use a different word than 'resulting'. The presence of antibodies targeting conserved epitopes only correlates with neutralization of non-vaccine parasites.

We replaced "resulting" with "leading to" to accurately reflect the observed increase in neutralization of non-vaccine parasites.

Line 88 – Perhaps change “and some AMA1 antibodies target domain 3” to “and some Inhibitory antibodies target domain 3”?

Done

Line 106 - Fusion-RD12 isn't in Suppl figure 1B or 2A and 2B and should be included if mentioned.

Thank you. Fusion-R_{DI2} is shown in Suppl Fig. 1A. We have now added SDS-PAGE and western blot data (Suppl Fig. 2C, 2D).

Line 130 – In figure 2D Box 3, how did you calculate the difference in the Beta-strand lengths? Is the main chain hydrogen-bonding pattern similar to the AMA-RON2L crystal structure? How did you calculate secondary structure? And how did you measure the 1.8 Å difference? The reason I ask is that they look very similar in the figure and Pymol doesn't always perfectly predict secondary structure.

The reviewer's point is well taken. Following further analysis of the backbone hydrogen bond architecture, we agree that there is likely insufficient evidence to warrant such a granular description with respect to absolute length and separation of the beta strands highlighted in the rightmost panel of Fig 2D. So as to not detract from this message, we have removed the rightmost panel in fig 2D and the associated text.

Line 139 – N136 or E136?

While the residues in loop 1e do not form crystal contacts in the fusion structure described herein, they do form crystal contacts in the 3ZWZ structure and this could explain some of the observed differences. To reflect this, we have included the following sentence,

“..., though it should be noted that differences in crystal packing between the fusion structure presented herein and the binary PfAMA1-RON2 structure may also contribute to observed shifts”

Line 141 - Do residues in loop 1e form crystal contacts in this structure or in 3zwz? Does that explain the difference in this loop? Are any of the differences due to differences in sequence between 3D7 and HP41?

Sequence alignment analysis reveals that there are no polymorphisms in loop e between the 3D7 and HP41 variants (Suppl Figure 1B).

Line 157 – group f?

This was a typo and was removed, thank you.

Suppl Figure 3B – Is there a data point missing in the AMA1D12 vs 3D7AMA1 group?

Thank you, we have updated the figure.

Line 181 – Are any of these differences in electrostatics due to the difference in sequence between 3D7 and HP41? For example, in loop 1bcd, there is an E to K polymorphism (Suppl Fig. 1).

A small number of electrostatic differences are due to polymorphisms between the 3D7

and HP41 sequences, though the only charge swap is the E to K polymorphism identified in the comment. Of the remaining 101 residues that comprise the loops, there are an additional 5 charged to neutral polymorphisms.

Line 189 - How did you calculate the ratio between EU(Fusion-FD12) vs EU(3D7AMA1) in Fig. 3D? It's unclear to me how it's possible to get a negative value.

The ratio between EU(Fusion-FD12) vs EU(3D7AMA1) was displayed as Log10 values and hence the negative values. This was done conservatively to show the variance in the data more clearly. We now present the ratio in linear scale as shown below (right side figure, Fig 3D).

Line 253 – Suppl Fig. 6C shows a greater proportion of IgM type antibodies against apoAMA1, not the fusion chimera, in the Fusion-FD12 immunized serum compared to apoAMA1D12. Is this correct or just mislabelled?

What our data suggests is that a greater proportion of IgM type antibodies in the Fusion-FD12 immunized animals bind the Fusion antigen compared to apoAMA1.

Suppl Fig. 2E – Change Q35 to Q359

Done

Reviewers' Comments:

Reviewer #1:

Remarks to the Author:

Thank you to the authors for the rebuttal to my previous remarks. Overall I'm satisfied that my previous concerns have been addressed and that this manuscript has been greatly improved.

There was one comment from the rebuttal that I could not observe had been addressed

Reviewer: "Fig 4A-C - the figure is titled as 'Proportion of IgG...' however it does not directly indicate the

proportion of IgG, for which values would fall between 0 and 1. These figures indicate the relative reactivity against the loop antigens for the same AMA1 titre."

Authors: "Thank you, we have changed "Proportion of IgG" to "Relative levels of IgG" to correctly reflect the assay."

When I look at the figure legend in the reviews document that I downloaded, these figures still appear to be titled as 'proportion of IgG...'"

On the same figure, I noticed that a different panel has been altered in the amended manuscript: figure 4E appears to have the same data as previously, but with a different Y scale (previously this was in the range 0-1. The axis units (EU FVOAMA1/EU 3D7AMA1) remain the same. The figure is still referred to as "Proportion of IgG binding to non-vaccine type FVOAMA1 682 to vaccine-type 3D7AMA1", however these negative values would appear to be a ratio, not proportion?

Reviewer #2:

Remarks to the Author:

All concerns have been addressed. Minor comments below.

Figure 4D - Please indicate on the figure somehow where RON2L binds. The polymorphisms are largely on one side. Is that the RON2L side or the reverse side? Does this help explain why the Fusion-FD12 immunogen elicits cross-strain specific Abs better than AMA1 alone?

Line 141 - Typo. N136 or E136?

Figure 5F and 5H are missing the P=.....

Response to reviewers

Reviewer #1 (Remarks to the Author):

Thank you to the authors for the rebuttal to my previous remarks. Overall I'm satisfied that my previous concerns have been addressed and that this manuscript has been greatly improved.

Thank you for the encouraging remarks and comments that helped improve our manuscript.

There was one comment from the rebuttal that I could not observed had been addressed

Reviewer: "Fig 4A-C - the figure is titled as 'Proportion of IgG...' however it does not directly indicate the

proportion of IgG, for which values would fall between 0 and 1. These figures indicate the relative reactivity against the loop antigens for the same AMA1 titre."

Authors: "Thank you, we have changed "Proportion of IgG" to "Relative levels of IgG" to correctly reflect the assay."

When I look at the figure legend in the reviews document that I downloaded, these figures still appear to be titles as 'proportion of IgG...'"

Thank you for pointing this out. We had made the change to the main document but missed it in the figure legend. We have now updated the figure legend with "Relative levels of IgG..."

On the same figure, I noticed that a different panel has been altered in the amended manuscript: figure 4E appears to the the same data as previously, but with a different Y scale (previously this was in the range 0-1. The axis units (EU FVOAMA1/EU 3D7AMA1) remain the same. The figure is still referred to as "Proportion of IgG binding to non-vaccine type FVOAMA1 682 to vaccine-type 3D7AMA1", however these negative values would appear to be a ratio, not proportion?

Yes, the data in Fig. 4E is the same, previously it was presented on a log scale, hence the negative values in y-axis. Following the reviewers comments we presented it in the linear scale, therefore the y-axis is positive.

Reviewer #2 (Remarks to the Author):

All concerns have been addressed. Minor comments below.

Thank you for the positive feedback and comments that helped improve our manuscript.

Figure 4D - Please indicate on the figure somehow where RON2L binds. The polymorphisms are largely on one side. Is that the RON2L side or the reverse side? Does this help explain why the Fusion-FD12 immunogen elicits cross-strain specific Abs better than AMA1 alone?

We added the following in the figure legend "RON2L binding site on AMA1 is indicated by an arrow".

The RON2L binding site on AMA1 is located at the top of the structure shown in Fig.4D. Most polymorphisms are located on loops surrounding the binding site and may explain at least in part the increase in broadly neutralizing antibodies induced by the fusion protein. But the exact nature of the broadly neutralizing antibodies targets remain to be seen.

Line 141 – Typo. N136 or E136?

Thank you, we have corrected this.

Figure 5F and 5H are missing the P=.....

Thank you, we have corrected this.